# Evaluation of Enterotoxins and Antimicrobial Resistance in Microorganisms Isolated from Raw Sheep Milk and Cheese: Ensuring the Microbiological Safety of These Products in Southern Brazil

**DOI:** 10.3390/microorganisms11061618

**Published:** 2023-06-20

**Authors:** Creciana M. Endres, Eliana Moreira, Andressa B. de Freitas, Andréia P. Dal Castel, Fábio Graciano, Michele B. Mann, Ana Paula G. Frazzon, Fabiana Q. Mayer, Jeverson Frazzon

**Affiliations:** 1Department of Food Science, Federal University of Rio Grande do Sul (UFRGS), Porto Alegre 91501-970, RS, Brazil; creciana.maria@gmail.com (C.M.E.); jeverson.frazzon@ufrgs.br (J.F.); 2SENAI/SC University Center, UniSENAI–Campus Blumenau, Blumenau 89036-256, SC, Brazil; 3SENAI/SC University Center, UniSENAI–Campus Chapecó, Chapecó 89813-000, SC, Brazil; moreira_eliana@hotmail.com; 4Institute of Food Technology, SENAI, Chapecó 89808-400, SC, Brazil; barellaandressa@gmail.com (A.B.d.F.); andreia.pdc2304@gmail.com (A.P.D.C.); 5Senior Field Application Specialist–Industry, BioMérieux Brasil SA, Indianópolis 04028-001, SP, Brazil; fabio.graciano@biomerieux.com; 6Department of Microbiology, Immunology and Parasitology, UFRGS, Porto Alegre 90010-150, RS, Brazil; mbertonimann@gmail.com (M.B.M.); ana.frazzon@ufrgs.br (A.P.G.F.); 7Department of Molecular Biology and Biotechnology, UFRGS, Porto Alegre 90010-150, RS, Brazil

**Keywords:** raw sheep’s milk, cheese, staphylococcal enterotoxin, microbiological safety, antimicrobial resistance

## Abstract

This study emphasizes the importance of monitoring the microbiological quality of animal products, such as raw sheep’s milk and cheese, to ensure food safety. In Brazil, there is currently no legislation governing the quality of sheep’s milk and its derivatives. Therefore, this study aimed to evaluate: (i) the hygienic-sanitary quality of raw sheep’s milk and cheese produced in southern Brazil; (ii) the presence of enterotoxins and *Staphylococcus* spp. in these products; and (iii) the susceptibility of the isolated *Staphylococcus* spp. to antimicrobial drugs and the presence of resistance genes. A total of 35 samples of sheep’s milk and cheese were examined. The microbiological quality and presence of enterotoxins were accessed using Petrifilm and VIDAS SET2 methods, respectively. Antimicrobial susceptibility tests were conducted using VITEK 2 equipment and the disc diffusion method. The presence of resistance genes *tet*(L), *sul*1, *sul*2, *erm*B, *tet*M, AAC(6)’, *tet*W, and *str*A were evaluated through PCR. In total, 39 *Staphylococcus* spp. were obtained. The resistance genes *tet*M, *erm*B, *str*A, *tet*L, *sul*1, AAC(6)’, and *sul*2 were detected in 82%, 59%, 36%, 28%, 23%, 3%, and 3% of isolates, respectively. The findings revealed that both raw sheep’s milk and cheese contained *Staphylococcus* spp. that exhibited resistance to antimicrobial drugs and harbored resistance genes. These results underscore the immediate need for specific legislation in Brazil to regulate the production and sale of these products.

## 1. Introduction

Sheep’s milk and cheese production in Brazil is a relatively new venture when compared to European countries. In 2017, the production of sheep’s milk in Brazil reached 1.72 million [1], with a significant portion being utilized for cheese production [2]. The microbiological quality of milk depends on its natural microbiota and potential contamination from viruses, bacteria, and fungi [3]. Therefore, it is crucial to assess the presence of microorganisms that indicate the cleanliness and safety of milk to prevent foodborne illnesses.

Foodborne illnesses are a major global health concern, impacting around 600 million individuals annually [4]. They occur when contaminated food is consumed and they can lead to serious consequences. *Staphylococcus aureus* is a pathogenic microorganism known for causing such illnesses, particularly through the production of enterotoxins. These toxins have been linked to outbreaks of foodborne diseases associated with cheese consuming of [4].

Cheese made from sheep’s milk is rich in proteins, fats, and carbohydrates, which often enhances the production of toxins by *S. aureus* [5]. The contamination of *S. aureus* often originates from human handling, water sources, milking equipment, and the environment [6]. *S. aureus* is known to cause mastitis in animals and is commonly found as a contaminant in raw milk [7]. Consequently, a key challenge in the dairy industry is to minimize the contamination levels in milk and its derivatives to ensure the safety of products and consumers [8]. Limited research has been conducted on the quality of raw sheep’s milk and the resulting cheese in Brazil, where specific legislation for these products is lacking.

Another significant concern in food safety is antimicrobial resistance (AMR), which results from misuse and abuse of antimicrobial agents in both humans and animals [9]. In 2019, antibiotic-resistant bacterial infections caused over 1.2 million deaths [10], with methicillin-resistant *S. aureus* (MRSA) alone accounting for 100,000 deaths and being a major cause of severe foodborne outbreaks [4]. Therefore, the objectives of this study were to assess: (i) the microbiological quality of raw sheep’s milk and cheese produced on farms in southern Brazil; (ii) the presence of *Staphylococcus* spp. and their enterotoxins in these samples; and (iii) the susceptibility of *Staphylococcus* spp. antimicrobial agents and the presence of resistance genes in these isolates.

## 2. Materials and Methods

### 2.1. Sampling and Experimental Design

Fifteen raw sheep’s milk samples were collected in three producing farms (F1, F2 and F3, n = 5 each) located in Rio Grande do Sul and Santa Catarina states (Figure 1). The selection of farms was based on convenience and their willingness to participate in the study. Additionally, 20 cheese samples from these farms including colonial, fresh, feta-type, and pecorino-type (n = 5 each) were purchased from the local market (Figure 1). All samples were within their specified expiration date and transported to the laboratory in Styrofoam boxes with ice and submitted to microbiological analysis under aseptic conditions within 24 h of collection. The samples were submitted to microbiological quality assessment through microbial counts of aerobic mesophilic microorganisms (AM), total coliforms (TC), *Escherichia coli*; and (d) *Staphylococcus aureus*. Moreover, enterotoxins’ analysis, *Staphylococcus* spp. isolation, antimicrobial susceptibility and resistance genes investigation were performed in 15 milk and 15 cheese samples. It is important to note that the fresh cheese samples were not included in these specific analyses.

### 2.2. Microbiological Quality Assessment

Microbiological analysis of aerobic mesophilic microorganisms (AM), total coliforms (TC), *Escherichia coli* and *S. aureus* were performed using the Petrifilm^TM^ system (3M Company, St, Paul, MN, USA). Each cheese sample (10 g) was mixed with 90 mL of buffered peptone saline in a sterile bag and homogenized in a stomacher for 5 min. Serial decimal dilutions were then performed in a 0.85% saline solution (up to 10^−4^) to enumerate the AM, TC, *E. coli* and *S. aureus*. For milk samples, 1 mL was inoculated into a tube containing 9 mL of 0.85% saline solution, followed with the serial dilutions (up to 10^−4^).

Petrifilm™ plates were used to quantify the counts of AM, TC, *E. coli*, and *S. aureus*. A 1.0 mL volume of each dilution was inoculated in the center of the lower film, and the upper film was carefully placed to prevent air bubble formation. Diffusers indicated for each plate type were employed to distribute the inoculum. The plates were then incubated for 24–48 h at 35 ± 1 °C, and the results were expressed in colony-forming units per milliliter CFU/mL [11].

### 2.3. Analysis of Staphylococcal Enterotoxins

Cheese samples (n = 15) and milk samples (n = 15) were evaluated for the presence of classical staphylococcal enterotoxins (SEs) using Vidas^®^ II staphylococcal enterotoxin (SET2, bioMérieux, Marcy l’Etoile, France), performed according to the manufacturer’s instructions. The methodology, based on the Enzyme Linked Fluorescent Assay (ELFA) technique using anti-SEs antibodies, allows the simultaneous detection of seven SEs serotypes (A, B, C 1, C 2, C 3, D, and E), with detection limit of 0.25 ng/g of food matrix. The system automatically quantifies the results as positive or negative.

### 2.4. Staphylococcus spp. Isolation

For each cheese sample, 25 g were placed in a sterile plastic bag with 225 mL of buffered peptone water. After homogenization in a mixer (Smasher™ AES Blue line) for 3 min, 1 mL of the initial 10^−1^ suspension divided into 0.3, 0.3, and 0.4 mL was seeded on 3 Baird-Parker agar plates (BP), supplemented with egg yolk tellurite emulsion.

Milk samples were inoculated directly into BP plates (0.3, 0.3 and 0.4 mL). Dilutions of up to 10^−3^ were performed for each sample. It was then incubated at 37 °C for 48 h. After 48 h, 5 typical colonies that appeared black, shiny, and convex and surrounded by zones of 2 to 5 mm and 5 atypical colonies were selected.

The isolates obtained were confirmed using Gram color to observe the morphology of the colonies. Subsequently, tests of catalase, coagulase and fermentation in mannitol salt were carried out. *Staphylococcus aureus* ATCC 25923 was used as a positive control in each of the biochemical test protocols.

The isolates were then seeded on TSA agar plates and incubated at 37 °C for 24 h. After this time, the colonies were suspended in a solution of 3 mL of 0.45% saline solution. A turbidity of 0.5–0.63 standard McFarland was established using VITEK DensiCHEK Plus (bioMérieux, Nürtingen, Germany). Isolates were identified to the species level using the VITEK 2 system (bioMérieux, Nürtingen, Germany) using GP cards (for analysis of gram-positive bacteria). The isolates identified as *Staphylococcus* spp. they were removed with a sterile loop from the TSA plate and placed in Eppendorf containing 1 mL of TSB broth with 10% glycerol and despaired at −20 °C.

### 2.5. Antimicrobial Susceptibility Analysis

The antibiotic resistance of *Staphylococcus* spp. isolates was assessed using the VITEK 2 method and the disk diffusion test using Mueller-Hinton agar (KASVI). A panel of 23 antimicrobial agents was examined. The antibiotics included in the VITEK 2 method were benzylpenicillin (BENPEN) oxacillin (OXA), cephalothin (CFL), cefovecin (CVN), ceftiofur (CEF), enrofloxacin (ENR), marbofloxacin, pradofloxacin (PRA), amoxicillin/clavulanic acid (AUG), kanamycin (K), gentamicin (GEN), neomycin (N), erythromycin (ERI), clidamycin (CLI), tetracycline (TET), doxycycline (DXT), chloramphenicol (CLO), nitrofurantoin (F) and Trimethoprim/Sulfamethoxazole (SXT), with concentrations established in the analysis cards. To perform the tests, 280 μL of standardized McFarland solution was combined with 3 mL of 0.45% saline solution. The VITEK 2 system (bioMérieux, Nürtingen, Germany) was used for antibiogram analysis with AST–GP80 cards [12]. For antibiotics not included in the AST-GP80 card, the disk-diffusion method on agar was conducted following the guidelines of the Clinical and Laboratory Standards Institute [13].

The antimicrobial agents tested by disc-diffusion method were selected based on those commonly used for treating *Staphylococcus* infection. The specific antimicrobials evaluated were ampicillin (AMP) (10 µg), linezolid (LNZ) (30 µg), rifampicin (RIF) (5 µg), sulfazotrim (SUT) (25 µg). *Staphylococcus aureus* strain ATCC25923 served as control. The isolates were classified as susceptible (S), intermediate (I), or resistant (R), according to Clinical and Laboratory Standards documents [13]. Multidrug resistance was defined as resistance to three or more antimicrobial classes.

### 2.6. Detection of Resistance Genes

*Staphylococcus* spp. DNA was extracted using PureLink Genomic DNA Mini Kit, according to the manufacturer’s instructions and described by Endres et al. 2021. The extracted DNA was stored at −20 °C until analysis. Polymerase Chain Reaction (PCR) was performed to detect the genes *erm*B, AAC(6)’, *tet*L, *tet*M, *tet*W, *sul*1, *sul*2, *str*A (Table 1). PCR for *16S rRNA* gene was used as internal control. Amplicons were submitted to electrophoresis using 1% agarose gels stained with ethidium bromide. Strains available in laboratory harboring the mentioned resistance genes were used as positive controls.

### 2.7. Statistical Analysis

The comparison of microorganisms’ counts was performed using the Kruskal-Wallis test. The different types of cheese were compared, and the raw sheep’s milk samples from the different farms. The significance level was set at *p* < 0.05. Afterwards, the Dunn test for multiple comparisons was conducted. Descriptive statistics were used to present the results enterotoxin detection, *Staphylococcus* spp. isolation, and resistance studies.

## 3. Results and Discussion

### 3.1. Microbiological Quality of Milk Samples

Out of the 15 milk samples obtained from the three farms, a significant occurrence of mesophilic microorganisms, total coliforms, *E. coli* and *S. aureus* was observed (Figure 2; Appendix A). Among analyzed samples, only one tested positive for the presence of enterotoxins. It was possible to isolate 24 strains of *Staphylococcu*s spp., identified as *S. sciuri*, *S. simulans*, and *S. aureus*. The isolated strains exhibited resistance to multiple antibiotics, and the presence of the resistance genes such as *tet*M, *sul*1, *Sul*2, *erm*B, *tet*M, and *str*A was detected (Appendix A).

There were no significant differences in the counts of total mesophilic aerobic microorganisms and *S. aureus* among the milk samples from the evaluated farms (Figure 2a,d). However, in terms of total coliforms, Farm 1 showed lower counts compared to Farms 2 and 3 (*p* = 0.022, and 0.027 respectively; Figure 2b). Farm 3 had higher counts of *E. coli* compared to Farms 1 and 2 (*p* = 0.013, and 0.031 respectively; Figure 2c). The differences in microorganism counts do not appear to be associated with geographic location, as Farms 1 and 3 are in Santa Catarina state and Farm 2 is in Rio Grande do Sul state. Other factors, such as hygiene practices during the milking process and milk storage on each farm, are likely to be involved. Moreover, the presence of *S. aureus* in raw milk is often associated with subclinical mastitis. However, no follow-up was conducted to assess the hygiene conditions or mastitis occurrence at the farms.

The presence of these microorganisms in milk can negatively impact its shelf life, quality, and risk of transmitting diseases, particularly when consumed raw or used to produce derivatives without proper pasteurization or the required maturation period as mandated by regulations [20]. Moreover, *S. aureus* has the potential to produce thermostable enterotoxins that can persist in the product even after pasteurization, posing additional risks to consumers [21]. Therefore, the results highlight the importance of microbial control and training for handlers.

### 3.2. Microbiological Quality of Cheese Samples

Cheese samples showed a lower occurrence of *E. coli* and *S. aureus*, and no presence of enterotoxins was detected. Out of the fifteen cheese samples analyzed, fifteen strains of *Staphylococcus* spp. were isolated, identified as: *S. lentus; S. warneri*, *S. pseudintermedius*, *S. chromogenes* and *S. sciuri*. The isolates demonstrated resistance to various antimicrobials and carried the genes *tet*L, *sul*1, *erm*B, *tet*M, *str*A, *AAC*(6), *str*A, *sul1*, *tet*W and *str*A (Appendix A).

Statistical differences in microorganism counts among different types of cheese were observed through Dunn’s test (Figure 3). These comparisons should be carefully considered, considering factors such as cheese production methods, milk composition, and maturation time specific to each cheese type. The analyzed cheese types undergo a distinct maturation process and are produced using pasteurized milk with added starter cultures, which shape final product’s microbiota [20]. Most of the cheeses analyzed exhibited low counts of total coliform, *E. coli*, and *S. aureus* (Figure 3; Appendix A). Colonial cheeses displayed lower total coliform counts compared to fresh cheeses (*p* < 0.05; Figure 3b), likely due to maturation process that hinders microorganism growth. There was no significant difference in *E. coli* counts observed between the different cheese types (Figure 3c). The low coliform counts indicate proper milk pasteurization and sound manufacturing practices.

On the other hand, all cheese samples showed significant mesophilic bacteria counts, with feta-type cheeses showing higher counts compared to colonial and fresh cheeses (Figure 3a). *S. aureus* was detected in fresh cheese but not in colonial and feta-type cheeses (Figure 3d). This finding may be attributed to factors such as the pH reduction, initial microbial load of the raw materials, and adherence to good manufacturing practices at the production sites. Moreover, the use of starter cultures and their byproducts play a crucial role in controlling pathogens during the maturation process [22].

*S. aureus* and *E. coli* are globally recognized as significant zoonotic pathogens that can lead to bacterial infections and foodborne illnesses [23]. These microorganisms have been detected in various food sources such as chicken meat, eggs, beef, sheep and goat meat, buffalo meat, raw bovine milk, raw sheep’s and goat’s milk, raw buffalo milk, cheese, and fish [24,25,26,27,28,29,30,31]. In Brazil, the Health Surveillance Secretariat reported 2504 outbreaks of foodborne diseases between 2016 and 2019, affecting 37,247 individuals and resulting in 38 deaths. However, it is important to note that the actual number of affected individuals may be higher as reporting foodborne outbreaks is not mandatory in Brazil [32]. Among these outbreaks, 11.5% were attributed to *S. aureus* as the causative agent, and 9.06% were associated with the consumption of milk and its derivatives [33]. Nonetheless, specific information regarding the frequency of staphylococcal food poisoning related to cheese is currently unavailable.

The Brazilian regulations for raw milk and dairy products (Ordinance 146 of 7 March 1996) set microbiological standards based on the moisture content of cheese. However, these regulations do not specifically address sheep’s milk and its dairy products, leaving a regulatory gap. The cheeses examined in this study have a moisture content ranging from 35.9% to 45.9%, classifying them as medium-moisture cheeses. The microbial counts observed in these cheeses complied with the established regulations, including the counts of total coliforms *S. aureus*. Only one sample of fresh cheese exceeded the acceptable level for thermotolerant coliforms (5.0 × 10^2^ CFU/mL).

Raw sheep’s milk generally exhibited higher microorganism counts compared to cheeses, which can be attributed to the use of pasteurized milk during cheese production and the maturation process. While the microbiological quality of cheeses is not solely dependent on milk pasteurization due to potential presence of thermoresistant toxins produced by certain microorganisms, pasteurization helps eliminate potential pathogens and spoilage bacteria present in raw milk [34]. Furthermore, maintaining sanitary conditions in the herd, following standard manufacturing process, reducing pH, removing water, and adding salt during cheese production favors microbiological safety [35].

### 3.3. Enterotoxin Investigation

In this study, a single sample of raw sheep’s milk was identified as containing staphylococcal enterotoxin, with the highest *S. aureus* count (5.30 × 10^3^ CFU/mL) observed using the traditional culture method. Previous studies have also reported the presence of enterotoxins in milk samples [36,37]. It is important to note that while this study focuses on characterizing *S. aureus*, both coagulase-positive (CoPS) and coagulase-negative (CoNS) Staphylococcus spp. possess genes that can produce enterotoxins [38,39]. Even though milk pasteurization, fermentation, and cheese maturation slow down the growth of *S. aureus*, it is crucial to examine the presence of enterotoxins, since they are the main responsible for staphylococcal food poisoning.

### 3.4. Staphylococcus spp. Isolation

A total of 461 colonies were from 15 samples of raw sheep’s milk and 15 samples of cheese. These colonies were analyzed using the catalase test. The negative catalase strains were discarded, and positive catalase strains (40%; n = 186) underwent the mannitol salt fermentation test and Gram staining. Out of these strains, 39 showed gram-positive staining in clusters and were identified as *Staphylococcus* spp. Of these isolates, 24 were from sheep’s milk and 15 from cheese. Among the cheese strains, eight (53%) were from feta-type cheese, four (27%) from pecorino-type cheese, and three (20%) from colonial cheese. Among the raw sheep’s milk strains, 13 (54%) were from Farm 1, 1 (4%) from Farm 2, and 10 (42%) from Farm 3. All the isolates were identified (Table 2), with the main *Staphylococcus* species in raw sheep’s milk being *S. sciuri* (66.7%), and the predominant species in cheese being *S. lentus* (60%). Most isolated strains were characterized as CoNS species (Table 2).

*Staphylococcus aureus* was detected exclusively in raw sheep’s milk samples. As mentioned above, cheese preparation includes steps that decrease the microbial count. The production of organic acids (such as lactic, acetic, propionic, and butyric acids) by *Lactobacillus* spp. during cheese ripening leads to lower pH [40], which inhibits bacterial growth, although certain pathogens can still survive in acidic conditions. However, there is still a risk of contamination during the cheese forming and packing processes.

Previous studies have assessed *Staphylococcus* spp. in sheep’s milk and identified *S. chromogenes*, *S. epidermidis*, *S. haemolyticus*, *S. pseudintermedius*, *S. aureus*, and *S. agnetis* [21], [41,42]. The variation in species profile could be attributed to factors such as the animals’ microbiota, management practices, and other factors not evaluated in this study.

*Staphylococcus pseudintermedius*, a coagulase-positive *Staphylococcus*, was detected in the cheese samples. It is commonly associated with infections in dogs, cats, and humans [43]. Among the coagulase-negative isolates, *Staphylococcus chromogenes* has been identified in bovine, sheep’s, and goat’s milk and is known to cause mastitis [44,45,46]. *S. sciuri* is another species linked to mastitis and is often described as methicillin-resistant [47,48]. *S. warneri*, commonly found in humans and animals, can cause meningitis, endocarditis, and septic arthritis in humans [49]. *S. warneri* has been isolated from fish, and antimicrobial resistance is a concern [50]. In a previous study conducted in northern Italy, CoNS (including *S. equorum*, *S. lentus*, *S. simulans*, *S. sciuri*, and *S. xylosus*) were detected in both raw milk and cheese [51]. CoNS are often present in fermented foods as part of the normal microbiota and can contribute positively to flavor and aroma development by producing proteolytic and lipolytic enzymes. CoNS also have good tolerance to salt and acidity, which makes them frequently present in cheeses made from sheep’s milk.

### 3.5. Antimicrobial Susceptibility Tests and the Detection of Resistance Genes

The antimicrobial susceptibility of the 39 *Staphylococcus* isolates was evaluated using 23 antibiotics. Results showed that 87% of the evaluated *Staphylococcus* strains were resistant to at least one antibiotic, with 46% exhibiting multidrug resistance. All isolates from raw sheep’s milk were susceptible to linezolid, sulfazotrim, gentamicin, and nitrofurantoin. However, a high frequency of AMR to oxacillin was observed (Figure 4). The isolates obtained from cheese samples showed lower AMR frequency compared to the milk isolates. The cheese isolates were susceptible to ampicillin, linezolid, sulfazotrim, amoxicillin/clavulanic acid, gentamicin, kanamycin, neomycin, doxycycline, chloramphenicol, and trimethoprim/sulfamethoxazole. The highest resistance frequency was observed against rifampicin (Figure 4), and resistance to up to 15 antimicrobials was detected in one *S. aureus* isolate (Figure 5).

Regarding the antimicrobial classes, all strains were susceptible to oxazolidinones. However, resistance to rifamycins, sulfonamides, β-lactams, macrolides, fluoroquinolones, lincosamides, and tetracyclines was observed in at least one *Staphylococcus* isolate (Figure 6).

All *S. aureus* isolates obtained from Farm 3 were found to be multidrug-resistant. This finding raises concerns as multidrug-resistant bacteria pose a significant global public health issue, causing approximately 700,000 deaths annually [52]. The resistance mechanisms are diverse and are determined by specific resistance genes. Of particular concern is the presence of these genes in mobile genetic elements, which can facilitate their horizontal transfer [53]. Consequently, even if not inherently pathogenic, the presence of resistant microorganisms can have an impact on public health by prolonging hospitalization periods and complicating treatment procedures [54].

Antimicrobial resistance and foodborne outbreaks are important considerations within the One Health framework. In Brazil, the Ministry of Health, Environment, and Agriculture collaboratively addresses this issue. The objective is to implement programs, policies, legislation, and research that promote effective communication and collaboration among different sectors. The aim is to achieve better public health outcomes, including monitoring of Food and Waterborne Diseases. The presence of antimicrobial resistance in foodborne pathogens is widely recognized as a significant biological risk, prompting concerns and extensive research. In Romania, a study focused on assessing cheeses made from raw cow’s and sheep’s milk revealed that 49% of *S. aureus* isolates exhibited multidrug resistance. The authors emphasize the increasing concern over antimicrobial resistance and recommend the establishment of an integrated surveillance system to monitor antimicrobial resistance throughout the entire food chain [55]. Another study investigated milk samples, semi-ready products, and finished cow products, revealing that 50% of the samples were contaminated with *S. aureus*, with some of them presenting AMR [56].

Antimicrobial resistance can undermine the effectiveness of clinical treatments, potentially leading them to failure. The concern regarding food consumption stems from the potential acquisition of resistance genes and microorganisms present in food, which can occur due to environmental contamination from sources such as water, air, soil, or manure. Moreover, the use of antibiotics in animal production contributes to the selection of resistant microorganisms. In this study, the three farms were surveyed regarding antibiotic use in animals and the presence of other animals in the production environment. Farm 1 reported no antibiotic use and had Maremano shepherd dogs on the property. Farm 2 reported using gentamicin and had no other animal species. Farm 3 reported using sulfonamides and terramycins and engaged in bovine milk production. This assessment corroborates that antibiotic use can promote resistance [57], along with the presence of other animals that can facilitate the transfer of microorganisms and resistance genes [58]. However, other unmeasured factors may also contribute to AMR, particularly in Farm 1, which claims no antibiotic use. It is important to note that cross-resistance, defined as resistance to antimicrobial agents that act through a shared pathway or target [59], can also occur.

This study also investigated some resistance genes for which the protocols and primers were available at the laboratory. The results revealed that *Staphylococcus* spp. isolates carried various resistance genes, irrespective of their pathogenicity (Figure 5). Among the isolates, 82% harbored the *Tet*M gene, 59% had *erm*B, 36% had *str*A, 28% had *tet*L, 23% had *sul*1, and 3% had *sul*2 and AAC(6)’. Notably, the *tet*W gene was not detected in any of the strains. The *tet*M gene, associated with tetracycline resistance, is commonly found on the Tn916 transposon, which can naturally be transferred between various Gram-positive and Gram-negative bacteria. The Tn916 transposon has been identified in coliform strains isolated from raw cow’s milk [60]. The *sul*1, *erm*B, *str*A, and *sul*2 genes have been described in *Salmonella* spp. isolated from leafy vegetables, chicken carcasses, and raw milk processing environments [61,62]. Studies have highlighted the potential risk of horizontal transfer of these genes between *Staphylococcus* strains, leading to the expansion and distribution of multidrug-resistant strains [63].

## 4. Conclusions

The study revealed that raw sheep milk had high levels of total coliform, *S. aureus*, and *E. coli*. However, most of cheese samples had low counts of these microorganisms. Enterotoxin was detected in one sample of raw sheep’s milk, which is concerning because enterotoxins can withstand heat and persist even after pasteurization, posing a risk to consumer health. The study also identified various species of *Staphylococcus* in both raw sheep’s milk and cheese. These isolates showed resistance to multiple antimicrobial agents, with 46% being multidrug-resistant. Additionally, most isolates carried at least one resistance gene, highlighting the alarming issue of antimicrobial resistance as a public health concern. These findings underscore the need for rigorous sanitary controls and responsible use of antimicrobials through regulations and monitoring by authorities. It is essential to establish guidelines for the production of sheep’s milk products, incorporating quality measures to ensure the safety of produced food. While this study provides initial insights into the quality of sheep milk and cheese samples, the antimicrobial resistance of *Staphylococcus* spp. strains associated with foodborne illnesses, and the presence of enterotoxins in southern Brazil, it is important to note that the number of samples analyzed was limited. Further studies are necessary to enhance understanding and contributions to regulatory bodies, manufacturers, and producers involved in the production of these products.

## Figures and Tables

**Figure 1 microorganisms-11-01618-f001:**
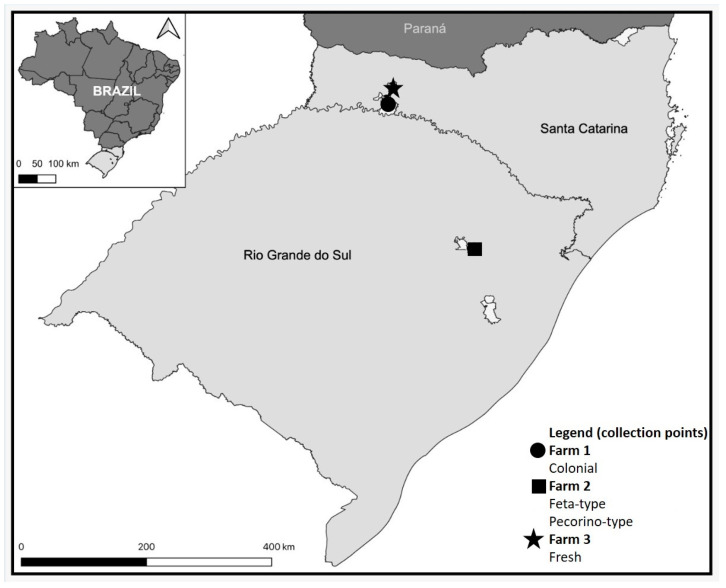
Sampling sites of sheep’s raw milk and cheese.

**Figure 2 microorganisms-11-01618-f002:**
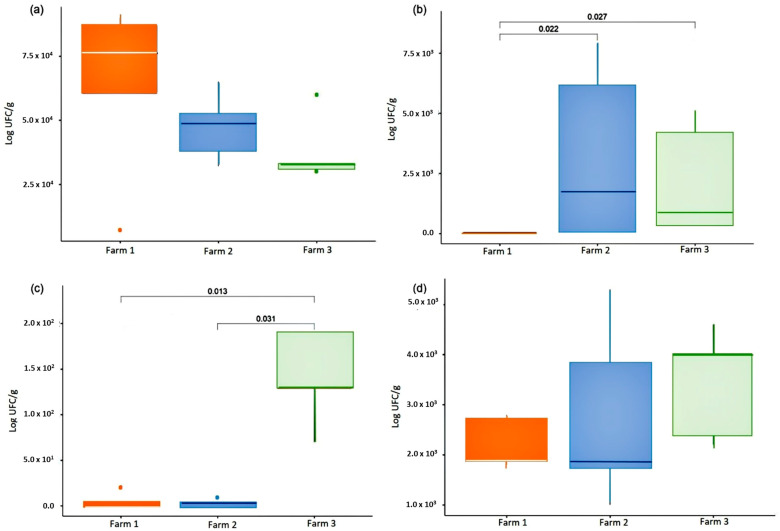
Microbiological quality of raw sheep’s milk. (**a**) Aerobic mesophilic microorganism counts (AM); (**b**) total coliforms (TC); (**c**) *Escherichia coli*; and (**d**) *Staphylococcus aure*us.

**Figure 3 microorganisms-11-01618-f003:**
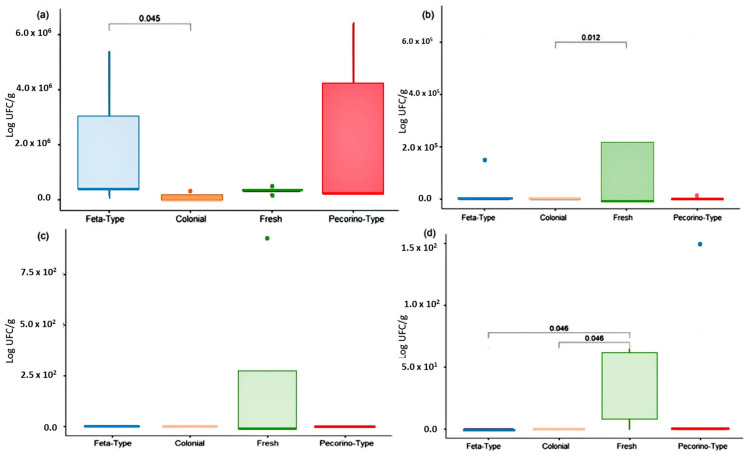
Microbiological quality of sheep’s cheese. (**a**) Aerobic mesophilic microorganism counts; (**b**) total coliforms; (**c**) *Escherichia coli*; and (**d**) *Staphylococcus aureus*.

**Figure 4 microorganisms-11-01618-f004:**
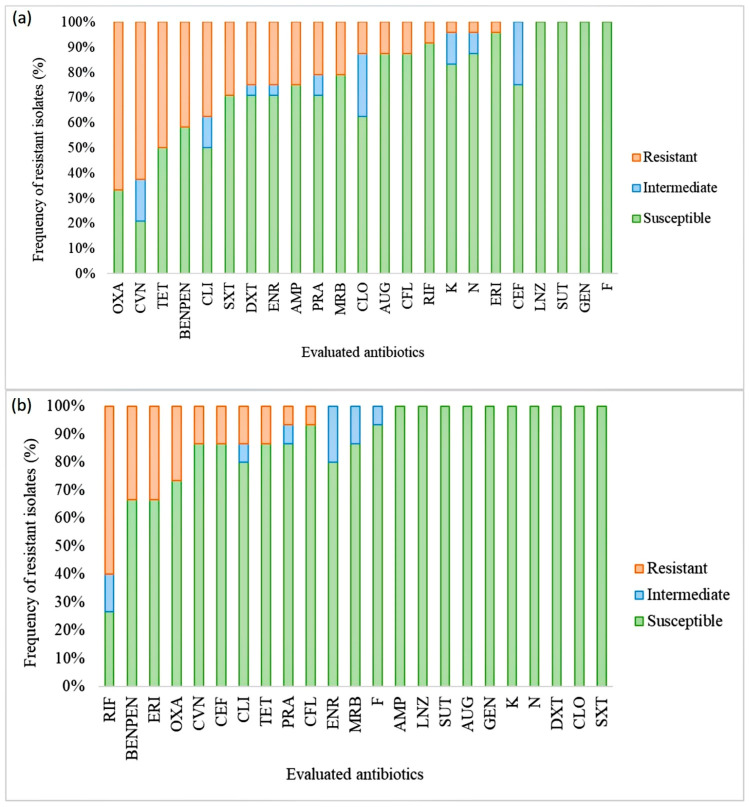
Antimicrobial susceptibility profile of *Staphylococcus* spp. isolates to different antimicrobials. Raw sheep milk isolates are shown in (**a**), and sheep cheese isolates in (**b**). Susceptible isolates did not show resistance to any class of antimicrobials; resistant isolates showed resistance to at least one class, and multidrug-resistant isolates showed resistance to 3 or more classes of antimicrobials.

**Figure 5 microorganisms-11-01618-f005:**
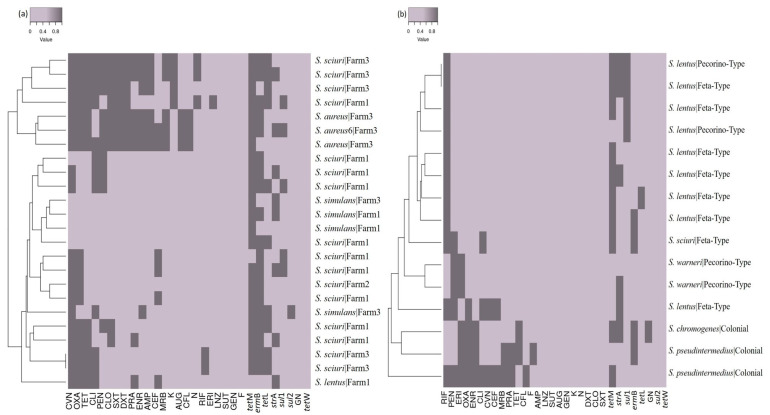
Presence/absence matrix and dendrogram cluster showing the antimicrobial susceptibility profile of each *Staphylococcus* spp. isolate and its resistance genes. Hierarchical groupings are presented for milk (**a**) and cheese (**b**) samples. The light gray represents antimicrobial susceptibility and the absence of investigated genes, while dark gray shows the presence of antimicrobial resistance and resistance genes.

**Figure 6 microorganisms-11-01618-f006:**
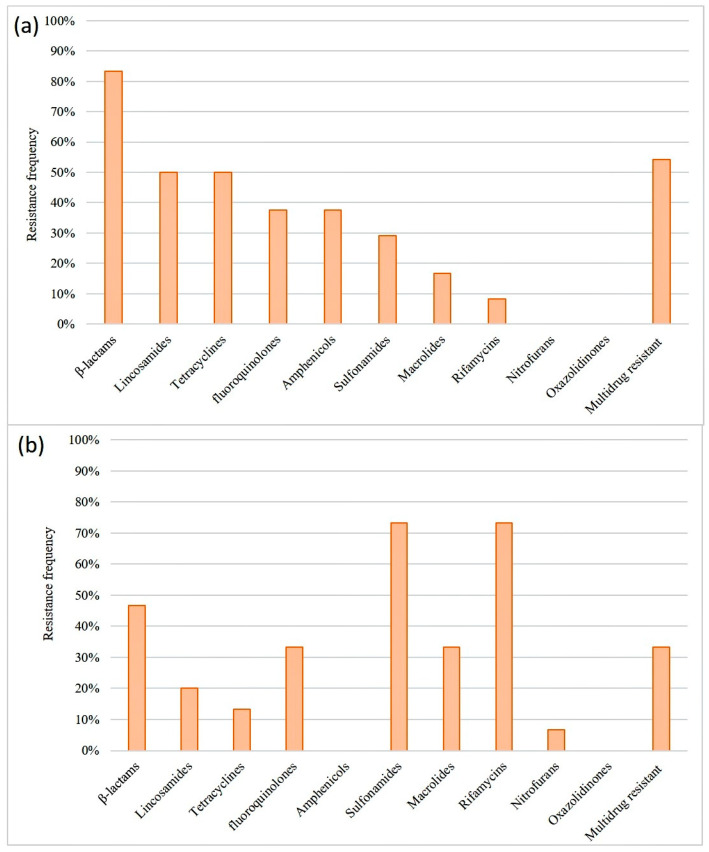
Frequency of *Staphylococcus* spp. resistance according to antimicrobial class. (**a**) Strains isolated from raw sheep’s milk. (**b**) Strains isolated from cheeses.

**Table 1 microorganisms-11-01618-t001:** Nucleotide sequences used as primers in PCR for identification of resistance genes and internal control.

Target	Annealing Temperature (°C)	Amplicon Size (bp)	Sequence (5′ to 3′)	Reference
*16S rRNA*	55	375	F CACGGTCGKCGGCGCCATT	[14]
			R GGACTACHVGGGTWTCTAAT	
*erm*B	52	639	F GAAAAGGTACTCAACCAAATA	[15]
			R AGTAACGGTACTTAAATTGTTTAC	
AAC(6)’	60	219	F CCAAGAGCAATAAGGGCATA	[16]
			R CACTATCATAACCACTACCG	
*tet*L	58	628	F ACTCGTAATGGTTGTAGTTGC	[16]
			R TGTAACTCCGATGTTTAACACG	
*tet*M	52	657	F GTTAAATAGTGTTCTTGGAG	[17]
			R CTAAGATATGGCTCTAACAA	
*tet*W	60	168	F GAGAGCCTGCTATATGCCAGC	[18]
			R GGGCGTATCCACAATGTTAAC	
*Sul*1	60	99	F GGATCAGACGTCGTGGATGT	[19]
			R GTCTAAGAGCGGCGCAATAC	
*Sul*2	57	99	F CGCAATGTGATCCATGATGT	[19]
			R GCGAAATCATCTGCCAAACT	
*str*A	59	99	F CCAGTTCTCTTCGGCGTTAG	[19]
			R ACTCTTCAATGCACGGGTCT	

**Table 2 microorganisms-11-01618-t002:** *Staphylococcus* species isolated in the present study.

Sample	Species	Frequency (%)	Coagulase Test	AMR Frequency (%) *
Milk	*S. sciuri*	16 (67)	CoNS	15 (94)
	*S. simulans*	4 (17)	CoNS	3 (75)
	*S. aureus*	3 (13)	CoPS	3 (100)
	*S. lentus*	1 (4)	CoNS	1 (100)
	Total	24 (100)		22 (92)
Cheese	*S. lentus*	9 (60)	CoNS	8 (89)
	*S. warneri*	2 (13)	CoNS	2 (100)
	*S. pseudintermedius*	2 (13)	CoPS	2 (100)
	*S. chromogenes*	1 (7)	CoNS	1 (100)
	*S. sciuri*	1 (7)	CoNS	1 (100)
	Total	15 (100)		14 (93)

* Resistant to at least one antimicrobial.

## Data Availability

The data presented in this study are available on request from the corresponding author.

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
