# Peer review of "Evaluation of Enterotoxins and Antimicrobial Resistance in Microorganisms Isolated from Raw Sheep Milk and Cheese: Ensuring the Microbiological Safety of These Products in Southern Brazil"

_microorganisms, 2023, doi:10.3390/microorganisms11061618_

Round 1

Reviewer 1 Report

The article entitled "Microbiological safety of raw sheep's milk and cheese produced in southern Brasil: an evaluation of enterotoxins and antimicrobial resistance in isolated Staphylococcus species " need to be strongly improved. First of all the title have to be changed because it suggests that the authors are focusing only on staphylococcal enterotoxins and antimicrobial resistance of isolated Staphylococcus species while there are also other bacteria described. The scheme of an experment is not well described, it should be clearly indicated how many colonies of Staphylococcus spp. were isolated from each sample. It is hard to imagine what was the idea if there was 30 samples and 461 of typical and atypical colonies while in the table with the results only 39 colonies are identified. What about the remaining over 400? There is very weak description of the analysis of staphylococcal enterotoxins using the Vidas SET2, it should be clearly shown which colonies have been chosen to do the Vidas procedure and how it was performed. It is shown that only one Vidas result was positive but out of what? The results are written together with the discussion but still you should start with the results and not the discussion because it is hard to find the results of the presented paper if they are already in the discussion. I would suggest to do a table to summarize all the results and to briefly describe it in the results. Also, some descriptions are written in few places in the text in a bit of different way. I suggest to reorganize the sections of a paper to make it clear to the readers what have been done, how and what are the results. Also, the references should be clearly checked because in some cases the cited fragments seem not to be from the indicated publications eg. line 240 ref. 21. Because of the above comments in my opinion the article need to be strongly improved and therefore it is no point to indicate smaller suggestions line by line.

I would suggest an english native speaker to check the quality of english used in the article.

Author Response

The article entitled "Microbiological safety of raw sheep's milk and cheese produced in southern Brasil: an evaluation of enterotoxins and antimicrobial resistance in isolated Staphylococcus species " need to be strongly improved.

First of all the title have to be changed because it suggests that the authors are focusing only on staphylococcal enterotoxins and antimicrobial resistance of isolated Staphylococcus species while there are also other bacteria described. 

Answer:  Thank you for the suggestion, the title of the article has been changed to “Evaluation of enterotoxins and antimicrobial resistance in microorganisms isolated from raw sheep milk and cheese: Ensur-ing the microbiological safety of these products in southern Brazil” leaving it broader without focusing on staphylococcal enterotoxins and antimicrobial resistance of isolated species of Staphylococcus.

The scheme of an experiment is not well described, it should be clearly indicated how many colonies of Staphylococcus spp. were isolated from each sample. It is hard to imagine what was the idea if there was 30 samples and 461 of typical and atypical colonies while in the table with the results only 39 colonies are identified. What about the remaining over 400?

Answer: Thank you for the suggestion, the requested information was added in the item 3.4. In addition to describing the number of initially isolated colonies, which were 461, we inserted the remaining data. Of these 461 colonies, only 186 were catalase positive. The 186 were submitted to the mannitol salt fermentation test and Gram staining,resulting in 39 colonies that were identified as gram positive and cocci grouping. The other colonies did not show Staphylococcus characteristics and were not included in the further analysis. Moreover, we have added a more detailed explanation on experimental procedures in section 2.1.

There is a very weak description of the analysis of staphylococcal enterotoxins using the Vidas SET2, it should be clearly shown which colonies have been chosen to do the Vidas procedure and how it was performed. It is shown that only one Vidas result was positive but out of what?

Answer: Thank you for your comment. We have added additional information on Vidas SET2 as requested.

The results are written together with the discussion but still you should start with the results and not the discussion because it is hard to find the results of the presented paper if they are already in the discussion. I would suggest to do a table to summarize all the results and to briefly describe it in the results. 

Answer: Thank you for your comment. We have changed the text in order to make it clearer, and we also have made the Table as suggested  (Supplementary Tables 1 and 2 of the revised version of the manuscript).

Also, some descriptions are written in few places in the text in a bit of different way. I suggest to reorganize the sections of a paper to make it clear to the readers what have been done, how and what are the results. 

Answer: Thank you for pointing out this improvement opportunity. We have checked and corrected the text. We hope that it has now a more fluid reading.

Also, the references should be clearly checked because in some cases the cited fragments seem not to be from the indicated publications eg. line 240 ref. 21. 

Answer: Thank you for pointing out this mistake. We have carefully checked the references.

Because of the above comments in my opinion the article need to be strongly improved and therefore it is no point to indicate smaller suggestions line by line.

Comments on the Quality of English Language:

I would suggest an english native speaker to check the quality of english used in the article.

Answer: We thank the reviewer for the time for reviewing the article and the valuable suggestions that in our opinion have improved it. The English language  was checked and edited. We hope that now the article is suitable for publication.

Reviewer 2 Report

This is an excellent well conducted and written study, with an original contribution to the knowledge of microbiological safety of raw sheep’s milk and cheese produced in southern Brazil: an evaluation of enterotoxins and antimicrobial resistance in isolated Staphylococcus species.

I encourage it’s acceptance after appropriate minor modifications as outlined below:

L70: “three producing farms” – Can you tell me which is the total sheep number in each farm. Regard to this question, I’m wondering why did you collect only 15 samples of raw sheep milk? Do you think that this number is suggestive for an objective result?

L162: “S. aureus” – Species names should be written in italic. Please ensure that the scientific name of all the mentioned species, throughout the manuscript, is written in italics.

L165: “E. coli” – Same aspect as the one mentioned before

L181: “S. aureus was observed...” I would like to suggest the authors to use use „was identified”. It is more appropriate.

L268: The documentation about the public health importance and awareness of the dramatically increasing antimicrobial resistance phenomenon of Staphylococcus aureus isolates over the last decade must be improved with some recently published valuable articles in Antibiotics (e.g. https://doi.org/10.3390/antibiotics10121458 and https://doi.org/10.3390/toxins8030062). These manuscript can be consulted and cited.

L309: L62: Regarding to this concern (just a recommendaon), you can add one or two phrases

about One Health concept and it’s status in Chile.

L62: Regarding to this concern (just a recommendaon), you can add one or two phrases

about One Health concept and it’s status in Chile.

L309: Regarding to this concern (just a recommendation), you can add one or two phrases

about One Health concept and it’s status in Brazil.

L358: Within the conclusion section, I would like to advise the authors to honestly underline the study limitations (if they exist), and mention further perspectives in the studied research area.

Author Response

This is an excellent well conducted and written study, with an original contribution to the knowledge of microbiological safety of raw sheep’s milk and cheese produced in southern Brazil: an evaluation of enterotoxins and antimicrobial resistance in isolated Staphylococcus species.

Answer: Thank you very much for the time for reviewing the article and the valuable comments that in our opinion have improved it.

I encourage it’s acceptance after appropriate minor modifications as outlined below:

 L70: “three producing farms” – Can you tell me which is the total sheep number in each farm. Regard to this question, I’m wondering why did you collect only 15 samples of raw sheep milk? Do you think that this number is suggestive for an objective result?

Answer: We thank you for pointing out this question, which was added in the conclusion section as a limitation. We have sampled those 3 farms by convenience, since their owners agreed to participate in the study (information now added in the section 2.1). The number of samples was defined for a previous study from our group (Endres et al., 2021), which accessed the milk and cheese microbiota through high throughput sequencing, an expensive method that limited the number of samples to be analyzed. The total number of sheep in each farm was around 350 animals.

L162: “S. aureus” – Species names should be written in italic. Please ensure that the scientific name of all the mentioned species, throughout the manuscript, is written in italics.

Answer: Thank you for pointing out this mistake. We have carefully checked the manuscript and made all scientific names in italics.

L165: “E. coli” – Same aspect as the one mentioned before

Answer: Thank you for pointing out this mistake, we have correct it.

L181: “S. aureus was observed...” I would like to suggest the authors to use use „was identified”. It is more appropriate.

Answer: Thank you for the suggestion, we have English checked and improved along all the manuscript.

L268: The documentation about the public health importance and awareness of the dramatically increasing antimicrobial resistance phenomenon of Staphylococcus aureus isolates over the last decade must be improved with some recently published valuable articles in Antibiotics (e.g. https://doi.org/10.3390/antibiotics10121458 and https://doi.org/10.3390/toxins8030062). These manuscript can be consulted and cited.

Answer:  Thank you for the comment. We inserted the two references in the discussion section as indicated and we understand that both have great relevance and contribution to the study.

L309: L62: Regarding to this concern (just a recommendaon), you can add one or two phrases about One Health concept and it’s status in Chile.

Answer: Thank you for the suggestion. Since the study was performed in Brazil, we have added information about this issue following the Brazilian reality on One Health programs.

L62: Regarding to this concern (just a recommendaon), you can add one or two phrases about One Health concept and it’s status in Chile.

Answer: Thank you for the suggestion. Since the study was performed in Brazil, we have added information about this issue following the Brazilian reality on One Health programs.

L309: Regarding to this concern (just a recommendation), you can add one or two phrases about One Health concept and it’s status in Brazil.

Answer: Thank you for the suggestion.We have added information about this issue following the Brazilian reality on One Health programs.

L358: Within the conclusion section, I would like to advise the authors to honestly underline the study limitations (if they exist), and mention further perspectives in the studied research area.

Answer: Thank you very much for the suggestion, which sometimes is a difficult exercise. We have added the limitations that we identified and study perspectives in the conclusion section. We thank again the reviewer and we hope that with the modifications made, the article is suitable for publication.